

# Classification and selection of the main features for the identification of toxicity in *Agaricus* and *Lepiota* with machine learning algorithms

Jacqueline S. Ortiz-Letechipia[1,*], Carlos E. Galvan-Tejada[1,*], Jorge I. Galván-Tejada[1], Manuel A. Soto-Murillo[1], Erika Acosta-Cruz[2], Hamurabi Gamboa-Rosales[1], José María Celaya Padilla[1] and Huizilopoztli Luna-García[1]

[1] Unidad Académica de Ingeniería Eléctrica, Universidad Autónoma de Zacatecas, Zacatecas, Zacatecas, México
[2] Departamento de Biotecnología., Universidad Autónoma de Coahuila, Saltillo, Coahuila, México
[*] These authors contributed equally to this work.

Corresponding author
Carlos E. Galvan-Tejada, ericgalvan@uaz.edu.mx

## ABSTRACT

The occurrence of fungi is cosmopolitan, and while some mushroom species are beneficial to human health, others can be toxic and cause illness problems. This study aimed to analyze the organoleptic, ecological, and morphological characteristics of a group of fungal specimens and identify the most significant features to develop models for fungal toxicity classification using genetic algorithms and LASSO regression. The results of the study indicated that odor, spore print color, and habitat were the most significant characteristics identified by the genetic algorithm GALGO. Meanwhile, odor, gill size, stalk shape, and twelve other features were the relevant characteristics identified by LASSO regression. The importance score of the odor variable was 99.99%, gill size obtained 73.7%, stalk shape scored 39.9%, and the remaining variables did not score higher than 18%. Logistic regression, k-nearest neighbor (KNN), and XG-Boost classification algorithms were used to develop models using the features selected by both GALGO and LASSO. The models were evaluated using sensitivity, specificity, and accuracy metrics. The models with the highest AUC values were XGBoost, with a maximum value of 0.99 using the features selected by LASSO, followed by KNN with a maximum value of 0.99. The GALGO selection resulted in a maximum AUC of 0.98 in KNN and XGBoost. The models developed in this study have the potential to aid in the accurate identification of toxic fungi, which can prevent health problems caused by their consumption.

# INTRODUCTION

The kingdom Fungi, which includes organisms commonly referred to as mushrooms, is composed of eukaryotic and heterotrophic organisms that exhibit a wide range of structures, functions, growth forms, and lifestyles (*Moreno, 703223 & Cvu, 2016*).
Fungi are present all over the world, impacting all habitats, constituting a diversity of approximately 1.5 million members worldwide (*Moreno, 703223 & Cvu, 2016*). Humans have made use of these organisms in various fields, such as medicine, bioremediation and food production. They are natural decomposers of dead matter, playing an important role in geochemical cycles by decomposing organic matter and producing nutrients for the soil, thus favoring plant growth (*Montoya, Arias & Agudelo, 2005*). One of the most significant human uses of mushrooms is their consumption as food. Humans are capable of consuming and assimilating certain species of mushrooms. Edible mushrooms are a rich source of protein, fiber, and amino acids. As a 100% vegetarian food, they are beneficial for individuals with diabetes, cardiovascular problems, and joint pain. Mushrooms do not contain cholesterol and assist in blood purification (*Rashed Khan, Shajun Nisha & Sathik, 2008*). Mushrooms that cannot be eaten in some cases causes diseases and serious poisoning that can be lethal, emphasizing the importance of proper classification. Additionally, certain mushrooms contain a high percentage of toxins, which can determine whether a mushroom is toxic or edible. Ingesting poisonous mushrooms can cause a disease called ''mycetism'', which can manifest in several types, depending on the characteristics of the poisoning. These include the cytotoxic (*White et al., 2019*), gyromitric (*White et al., 2019*), hallucinogenic, muscarinic, gastrointestinal, coprinic, and paxilic syndromes, among others. The most dangerous ingestions are the cytotoxic syndromes, which are often fatal (*Lazo, 1986*). Therefore, timely diagnosis is crucial in case of intoxication (*Ventura Pedret et al., 2020*). Within approximately 12 h, mushroom poisoning may present with diarrhea and vomiting. If the mushroom poisoning has a duration of less than four hours it is called short latency, on the contrary, if it lasts more than four hours it is long latency. Once consumed the uncontrolled wild specimen, within the latency period of 8-12-18 h, the presence of symptoms should be observed, even when the latency is shorter there is a risk of cytotoxic intoxication, taking into account that certain factors, such as the consumption of large meals of mushrooms rich in chitin or mixed meals, can shorten the latency period and mask the symptoms of amatoxin intoxication. In any presence of vomiting and diarrhea after mushroom consumption should be viewed with suspicion. In the absence of an expert who can identify the species of the specimen within 30 min after consumption, specific treatment for cytotoxic poisoning should be initiated. Methods to detect cytotoxins are either macroscopic for cytotoxin detection or microscopic for identification, and urinalysis is crucial. Once treatment is initiated before analysis, mortality rates can be as low as when treatment is initiated before analysis, mortality rates can be as low as 5% (*Flammer & Schenk-Jäger, 2009*).

This research proposes the identification of mushrooms through a relationship with artificial intelligence. This automatic classification has been involved in recent studies, such as ''A deep learning-based approach to edible, inedible and poisonous mushroom classification''. *Zahan et al. (2021)* demonstrates the use of AI for mushroom classification, the researchers utilized InceptionV3, VGG16, and Resnet50 techniques to categorize mushrooms in 8190 mushroom images into edible, inedible, and poisonous categories, with an 8:2 ratio of training and test data. The contrast limited adaptive histogram equalization (CLAHE) method was employed along with InceptionV3 to achieve the

highest test accuracy. A comparison was made between contrast-enhanced and non-contrast-enhanced methods. InceptionV3 achieved the highest accuracy of 88.40% among the other algorithms. Additionally, *Leichtmann et al. (2023)* tested a high-risk mushroom hunting task, in which an AI-based application was used to suggest classifications based on mushroom images to users, aiding them in deciding whether a mushroom specimen was edible or poisonous.

Some disadvantages of the aforementioned studies are the absence of feature selection and the computational cost that this implies in the prediction models, taking into account that the time that the person will spend for the analysis of 22 features will be quite a lot. To complement this research, data mining will be used to analyze the data, the performance comprises a set of techniques and technologies that enable the exploration of large databases, either automatically or semi-automatically, to identify repetitive patterns that explain the behavior of the data (*Ballesteros et al., 2018*), may also be defined as the practice of examining a pre-existing database to generate new information (*Osman, 2019*). The DM technique to be used is classification, based on a supervised machine learning approach used to predict group membership (*Kesavaraj & Sukumaran, 2013*), this technique makes predictions about data values using known results, this can be a resolution to ML problems (*Baradwaj & Pal, 2012*). The central aim of this study is to harness data mining (DM) and artificial intelligence (AI) techniques to meticulously identify and optimize the essential organoleptic, ecological, and morphological characteristics of fungi, specifically within the *Agaricus* and *Lepiota* genera, in relation to their implications for human toxicity. To achieve this, the study make use of renowned machine learning models, including logistic regression, GALGO, and LASSO Regression. Beyond simple classification, this research places a strong emphasis on the rigorous optimization of the classification protocol to augment its efficiency and reduce computational burdens.

## MATERIALS AND METHODS

This section focuses on the materials and the methodology used for the identification of poisonous mushrooms according to the morphological and organoleptic characteristics of a group of specimens. For a better understanding, the methodology is divided into six phases, presented from the first to the last to the last step for the identification of poisonous mushrooms, which are shown in Fig. 1. Figure 1A provides details about the data acquisition process, which involved downloading the mushroom dataset. Figure 1B elaborates on the data preprocessing steps, which included converting the data into numerical format. Figure 1C highlights the use of two feature selection techniques, GALGO and LASSO. Figure 1D outlines the implementation of the logistic regression classification algorithms, XGBoost, and KNN. Figure 1E validate with Cross Validation. Finally, Fig. 1F evaluates the results in terms of AUC, sensitivity, and specificity.

### Data acquisition

The dataset used in this study is presented by the Machine Learning Repository of the "Center for Machine Learning and Intelligent Systems", and it is denominated "Fungi Data Set" (*UCI Machine Learning Repository, 1987*). The dataset encompasses the family

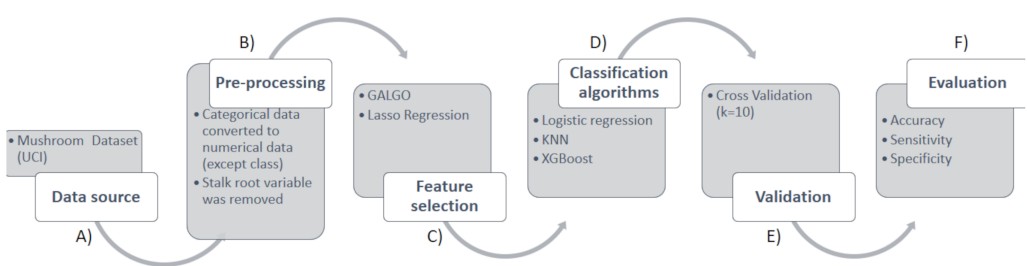

**Figure 1 Methodology for the prediction of toxicity in mushrooms.**

Agaricaceae (*Agaricus*, *Lepiota*, and allied species) as described in *Schlimmer (1981)* and *Dua & Graff (2017)*. The dataset is well-balanced, with 52% of the mushrooms being poisonous and 48% being edible. Importantly, the classes in this database consist of categorical data and refer to atemporal morphological characteristics, thus the year of publication of the database does not affect the results of the experiments. The three classes in the database correspond to mushrooms that are definitely edible, definitely poisonous, or of unknown edibility and not recommended, with the last class being combined with the poisonous class for the purposes of this study (*Schlimmer, 1981*). The following section outlines the preprocessing and feature selection steps that were performed on this dataset for the classification of the mushrooms into their respective classes. The 23 characteristics are: cap-shape, cap-surface, cap-color, bruises, odor, gill-attachment, gill-spacing, gill-size, gill-color, stalk-shape, stalk-root, stalk-surface-above-ring, stalk-surface-below-ring, stalk-color-above-ring, stalk-color-below-ring, veil-type, veil-color, ring-number, ring-type, spore-print-color, population and habitat.

## Pre-processing

In this study, data pre-processing was performed as the first step in Machine Learning to transform and encode the data for rapid parsing by the machine. It was found that data pre-processing is the most influential factor in the performance of supervised machine learning algorithms (*Maharana, Mondal & Nemade, 2022*). As all the attributes and observations in the database were categorical data, in order to evaluate the classifiers proposed in this work, categorical outputs were converted to numerical values. Furthermore, to enable a blind test, the data set was divided into two balanced subsets: the blind test set (25%) and the training set (75%). The database did not present unavailable data (na), so the next step was to convert the categorical values to numerical values (except for the class). Recalling that the database is balanced, therefore it is not a problem for data processing and no resampling was created. The stalk root variable was not recognized in approximately 70% of the specimens; hence, it was removed from the dataset. This could indicate that the characteristic is difficult to obtain or may have less influence on the classification results.

### *Feature selection and classification experiment*

The classification experiment is performed with the Integrated Development Environment (IDE) for the R programming language R-Studio. LASSO regression and GALGO (*Trevino*

*& Falciani, 2006*) were used as feature selection methods. Two feature selections were chosen for comparison, from selection by logistic regression with shrinkage tendency, which can be useful for simple sparse models, to selection by genetic models, ideal for multivariate models designed primarily to develop statistical models from large-scale data sets. GALGO is an object-oriented implementation of the genetic algorithm (GA) under the R language. It selects models with high fitness values using a GA procedure and implements functions for analyzing the selected model populations, as well as functions for reconstructing and characterizing representative summary models (*Li et al., 2001*).

LASSO (Least Absolute Shrinkage and Selection Operator) regression (*Tibshirani, 1996*) is a linear regression method used to identify the most important variables and corresponding regression coefficients in a model that minimizes prediction error. LASSO regression introduces a constraint on the model parameters that "shrinks" the regression coefficients towards zero, forcing the sum of the absolute value of the regression coefficients to be less than a fixed value λ. As a result, LASSO regression not only selects the important variables but also performs variable shrinkage and can be used for variable selection and regularization (*Ranstam & Cook, 2018*).

The proposed algorithms for classification in this study are logistic regression, k-nearest neighbor (KNN), and XGBoost. Three methods were chosen to obtain a multiple comparison. Starting with logistic regression, an uncomplicated regression analysis that provides the probability of an event occurring, ideal for outputting a binary target variable. KNN is a "lazy learning" classification model that uses proximity to make classifications or predictions about the clustering of an individual data point. XGBoost as an already optimized method for classification and regression, ideal for large data sets or data sets that present a mixture of categorical and numerical varieties and simple to use. This selection of methods could be useful for the implementation of models for biologists as its application is uncomplicated. Logistic regression is commonly used in situations where the aim is to predict the presence or absence of a feature or outcome based on the values of a set of predictors. It is similar to linear regression, but is adapted for models where the dependent variable is dichotomous. Logistic regression coefficients can be used to estimate the odds ratio of each independent variable in the model (*IBM, 2023*).

KNN classification works by classifying examples according to the class of their nearest neighbors. It is a simple technique that is often used when the training examples need to be in memory at runtime. The technique is more commonly known as k-nearest neighbor (KNN) Classification, where the KNNs are used to determine the class (*Cunningham & Delany, 2021*).

The XGBoost algorithm is a popular ensemble method used for both classification and regression problems. It is based on a sequential assembly of decision trees, specifically CART (Classification and Regression Trees) models. The trees are added sequentially in order to learn from the results of the previous trees and correct the error produced by them. This process continues until the error cannot be corrected any further, which is known as the "down-gradient" process. The XGBoost algorithm has been shown to be highly effective in many machine learning applications and has been widely adopted in both academia and industry (*Espinosa-Zúñiga & Espinosa-Zúñiga, 2020*).

*Validation*

Cross-validation is a popular resampling method that is used to estimate the true prediction error of models and adjust model parameters in order to avoid overfitting. This technique is widely used in machine learning and statistical modeling. K-fold cross-validation is a variant of this method where the data set is divided into k subsets, and then the model is trained iteratively by using some of these subsets, while the remaining subsets are used to evaluate its performance. This method is particularly useful for estimating the accuracy of a model and for selecting optimal hyperparameters for the model (*Hastie, Friedman & Tibshirani, 2001*; *Duda, Hart & Stork, 2001*; *Berrar, Lopes & Dubitzky, 2019*).

The evaluation metrics for predictive models, specifically for binary classifiers, include receiver operating characteristic (ROC) curve, sensitivity, and specificity. The area under the ROC curve (AUC) is a standard measure of the discriminative ability of a test to determine whether a particular condition is present or absent. An AUC value of 0.5 implies that the test has no discriminative ability, while an AUC value of 1.0 represents a test with perfect discrimination (*Hoo, Candlish & Teare, 2017*).

Sensitivity (SN) is a metric used to evaluate the ability of a test to correctly identify individuals with a positive disease status, such as poisonous mushrooms (*Hanley & McNeil, 1982*). It is calculated using Eq. (1), where PPV denotes positive predictive values, TP represents the number of true positives, and FP indicates the number of false positives (*Berrar, Lopes & Dubitzky, 2019*).

$$PPV = \frac{TP}{TP + FP}. \tag{1}$$

The specificity (SP) is a metric used to evaluate the ability of a test to accurately classify an individual as disease-free (*e.g.*, edible mushrooms). The calculation of SP is based on Eq. (2), where NPV stands for negative predictive values, TN represents the number of true negatives, and FN indicates the number of false negatives. The concept of SP is the opposite of sensitivity (SN), which refers to the ability of a test to correctly identify an individual as positive in disease (*e.g.*, poisonous mushrooms) (*Hanley & McNeil, 1982*). These metrics are important in evaluating the performance of binary classification models.

$$NPV = \frac{TN}{TN + FN}. \tag{2}$$

## EXPERIMENTS AND RESULTS

The mushroom specimens were classified as either toxic or edible based on their distinct characteristics. The results of the classification are presented below.

### LASSO

The results of the selection of the most significant characteristics by LASSO are presented in Fig. 2. An importance of 99.9% was obtained for the odor variable, 73.7% for the gill size, and 39.9% for the stalk shape. The remaining variables do not reach more than 18%. Therefore, it can be deduced that the odor, gill size, and stalk shape are the three
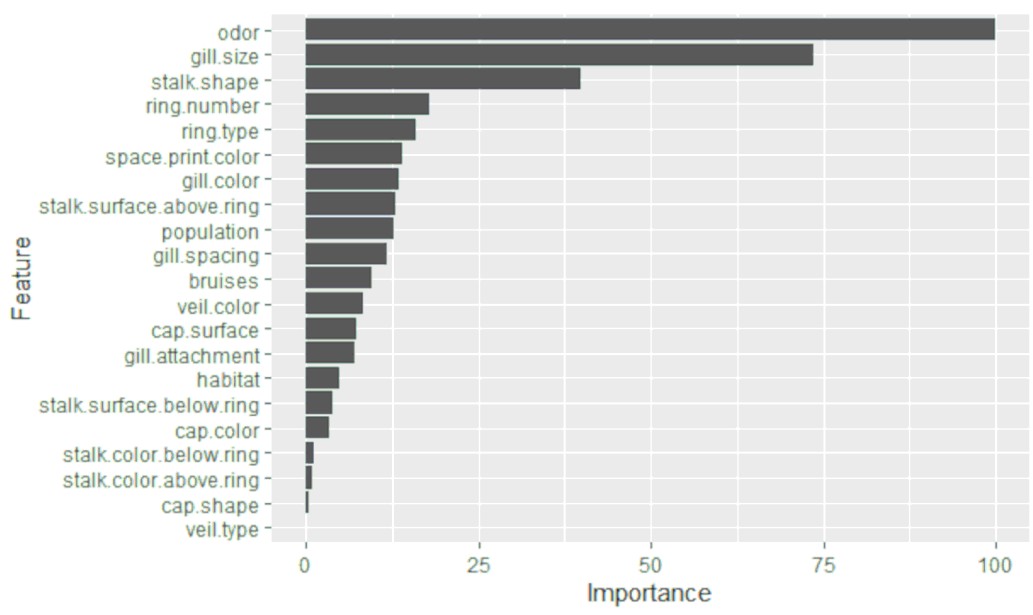

**Figure 2** **The importance of features obtained from LASSO selection is presented in a bar graph.** The *y*-axis represents each of the features, while the *x*-axis indicates the level of importance, scaled from 0 to 100.

most significant characteristics for the detection of toxicity in *Agaricus* and *Lepiota*. After performing LASSO analysis, 15 significant features were obtained. These features include cap.surface, cap.color, odor, gill attachment, gill spacing, gill size, gill color, stalk shape, stalk surface above ring, veil color, ring number, ring type, spore print color, population, and habitat.

## GALGO

The parameters utilized for the GALGO selection process were as follows: goal fitness of 1, maximum big bangs of 60, maximum number of generations of 60, and chromosome size of 5. These parameters were utilized in conjunction with the KNN classification method, which was chosen based on the current state of the art. In total, sixty models were generated with the aim of achieving a goal fitness closer to 1. From the resulting models, a ranking chart was generated to illustrate the frequency of the designated variables, with colors ranging from black (most frequent) to white (least frequent). The objective of this process was to create an optimal model. The GALGO feature selection method resulted in the identification of the three most significant characteristics for the classification of poisonous fungi, namely odor, spore print color, and habitat, which are highlighted in black in Fig. 3. On the other hand, gill attachment, stalk shape, and veil type were found to be insignificant and are shown in gray on the right side of Fig. 3.

In Fig. 4, the three most significant variables are indicated by highlighting their names on the right side of their respective frequency bars. The variables are presented in order of importance, with odor being the most significant, followed by spore print color and habitat.
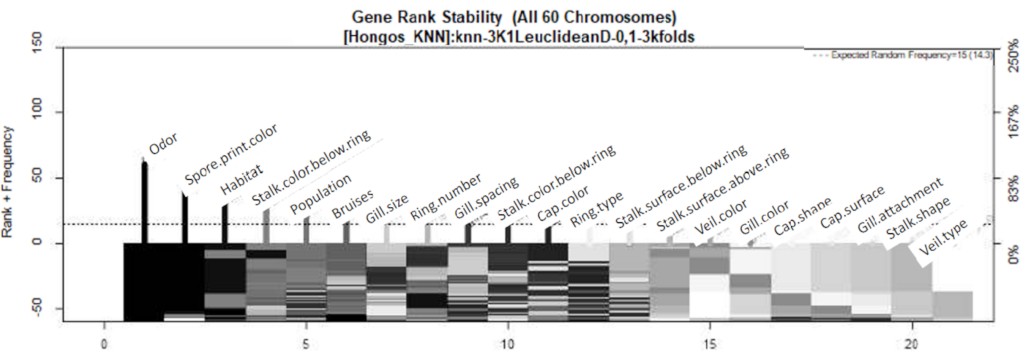

**Figure 3  Feature–rank stability in 60 models, for the classification of mushrooms, where the "negative y" axis shows the color-coded range based on color for each of the custom features in terms of the generation of each model.** The "x" axis represents the features provided by rank. The base color of each feature is designated based on its descending rank, from black to white.

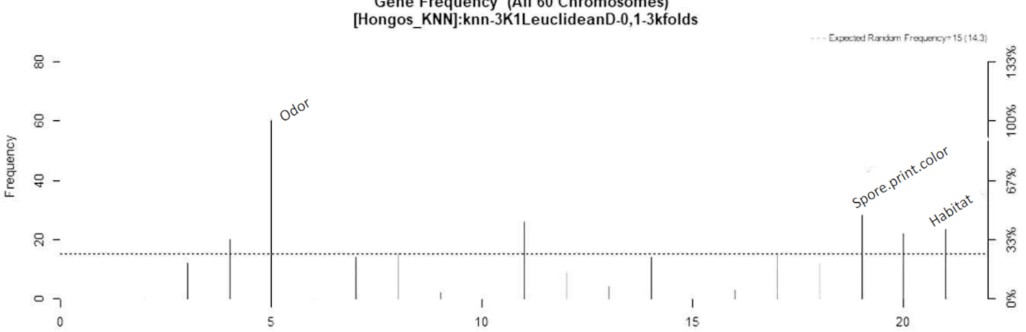

**Figure 4  Feature–rank stability in 60 models, for the classification of mushrooms, where the "negative y" axis shows the color-coded range based on color for each of the custom features in terms of the generation of each model.** The "x" axis represents the features provided by rank. The base color of each feature is designated based on its descending rank, from black to white.

## Validation

To validate the predictive models, cross-validation was utilized, with the total dataset partitioned randomly into a subset of 75% for model training and 25% reserved for blind testing. The process was repeated 10 times ($k = 10$) to avoid any bias in the predictions. The results of the classification models generated with the LASSO selected features are shown in Table 1. The logistic regression model yielded an AUC of 0.977, KNN showed an AUC of 0.989, and XGBoost measured an AUC of 0.999. The use of LASSO regression allowed for the reduction of the original 22 features to 15, which in turn reduced the computational cost and processing time, while maintaining the performance of the classifiers.

After applying the GALGO feature selection process, the highest performing classification model with an AUC of 0.999 was obtained using XG-Boost, as presented in Table 2. In contrast, logistic regression did not achieve similarly favorable values.
**Table 1   AUC, specificity and sensitivity of the models generated with the features selected by LASSO.**

|  | AUC | Sensitivity | Specificity |
|---|---|---|---|
| **XGBoost** | 0.999 | 1.0 | 1.0 |
| **KNN** | 0.989 | 0.976 | 0.937 |
| **Logistic regression** | 0.977 | 0.960 | 0.951 |

**Table 2   AUC, specificity and sensitivity of the models generated with the features selected by GALGO.**

|  | AUC | Sensitivity | Specificity |
|---|---|---|---|
| **XGBoost** | 0.999 | 1.0 | 0.996 |
| **KNN** | 0.998 | 0.989 | 0.987 |
| **Logistic regression** | 0.736 | 0.702 | 0.573 |

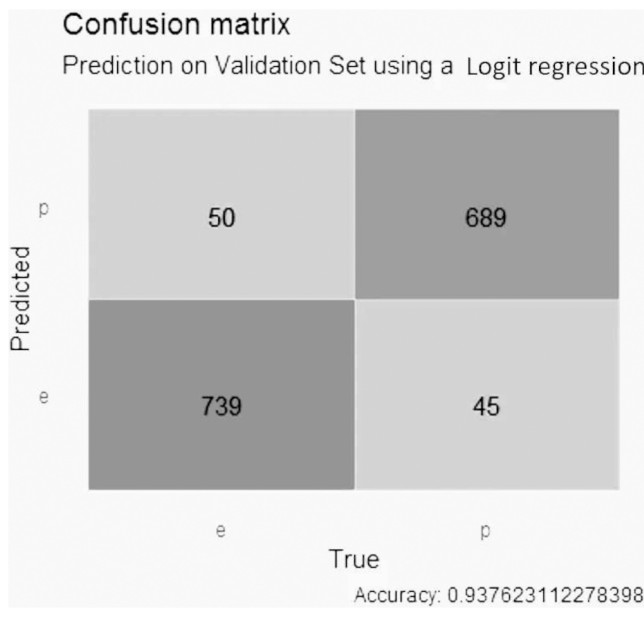

**Figure 5   Confusion matrix of the logistic regression model using LASSO selection.**

Based on the results obtained, the confusion matrix of the three predictive models concerning their feature selection method was created. For the logistic regression model with Lasso selection, shown in Fig. 5, the precision was close to 0.94, with 50 false positive values, 45 false negative values, 739 true negative values and 689 true positive values.

The KNN model with LASSO selection, in Fig. 6, obtained an accuracy of better than 0.99, with 1 false positive value, 0 false negative values, 788 true negative values and 734 true positive values.

In the XGBoost model with Lasso selection, shown in Fig. 7, the accuracy is perfect with a score of 1, with 0 false positive values, 0 false negative values, 789 true negative values and 734 true positive values.

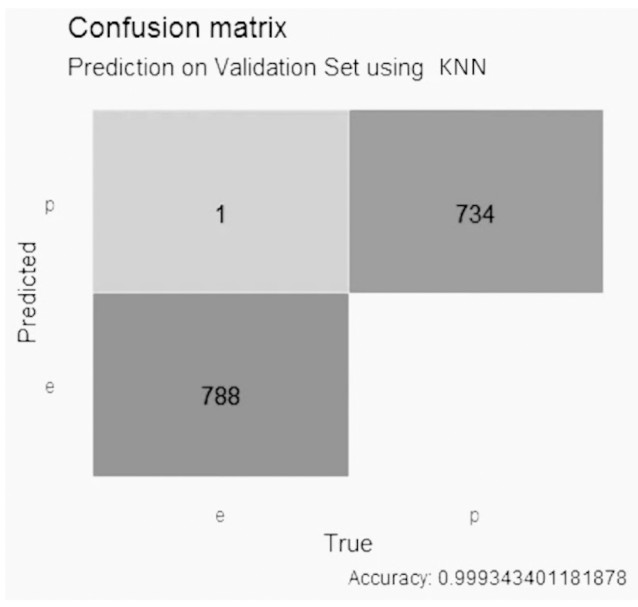

Accuracy: 0.999343401181878

**Figure 6** **Confusion matrix of the KNN model using LASSO selection.**

For the GALGO selection (Fig. 8), the logistic regression model obtained an accuracy close to 0.64, with 256 false positive values, 299 false negative values, 534 true negative values and 435 true positive values.

In the KNN model with GALGO selection, shown in Fig. 9, it presented an accuracy higher than 0.99, with two false positive values, one false negative value, 787 true negative values and 733 true positive values.

Figure 10 indicates that the XGBoost model using GALGO selection presented an accuracy higher than 0.99, with two false positive values, one false negative value, 787 true negative values and 733 true positive values.

## DISCUSSION AND CONCLUSION

In the study, LASSO and GALGO were employed to select the most significant characteristics for the detection of toxicity in mushrooms. LASSO returned odor, gill size, and stalk shape as the three most significant characteristics, with odor having 99.99% importance, gill size 73.7%, and stalk shape 39.9%. On the other hand, GALGO selection ranked odor, spore print color, and habitat as the most significant. Notably, odor was found to be the most relevant characteristic identified by both selection methods. However, it was highlighted that odor can be a subjective characteristic and difficult to differentiate for people who do not frequently participate in mushroom hunting. There have been numerous studies that have implemented different approaches to identify whether a mushroom is edible or poisonous. *Rashed Khan, Shajun Nisha & Sathik (2008)* compared several algorithms for mushroom classification, including the database used in this study and the 22 features. The models they created used expectation maximization (EM), an iterative method for finding maximum likelihood or maximum a posteriori (MAP)
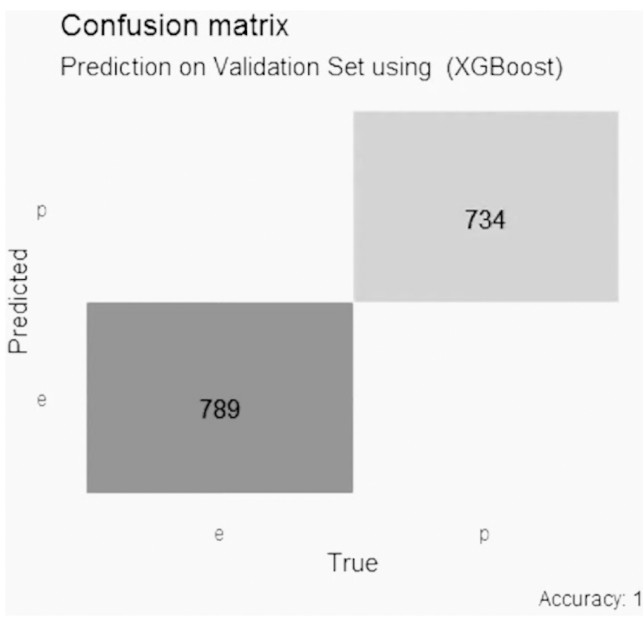

**Figure 7 Confusion matrix of the XGBoost model using LASSO selection.**

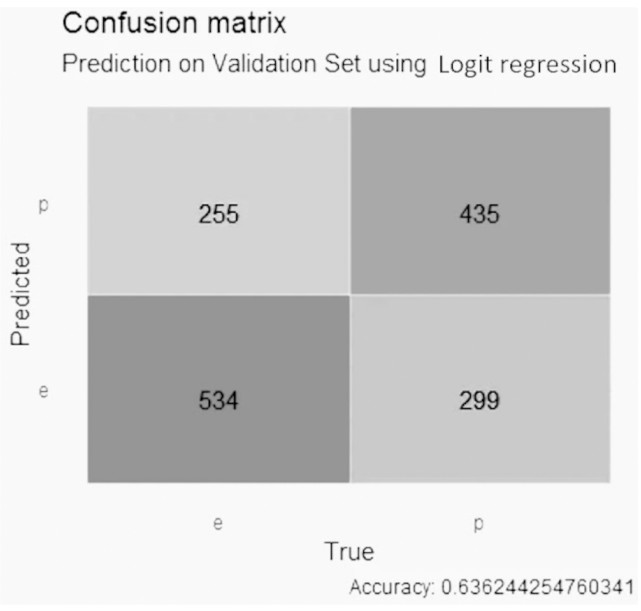

**Figure 8 Confusion matrix of the logistic regression model using GALGO selection.**

estimates of parameters in statistical models. Compared to the time required to build the model, the faster algorithm takes much less time than EM and K-means. The K-means algorithm proved to be the best algorithm in terms of correctly clustered instances and expectation maximization for this mushroom data set. The results for correctly clustered
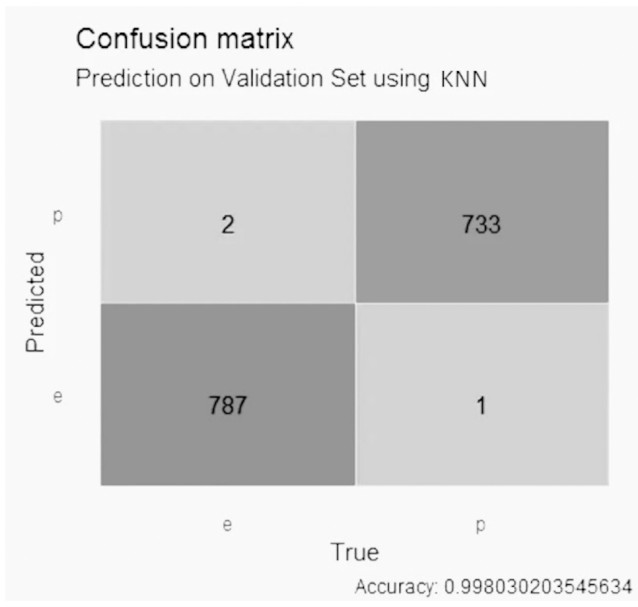

**Figure 9 Confusion matrix of the KNN model using GALGO selection.**

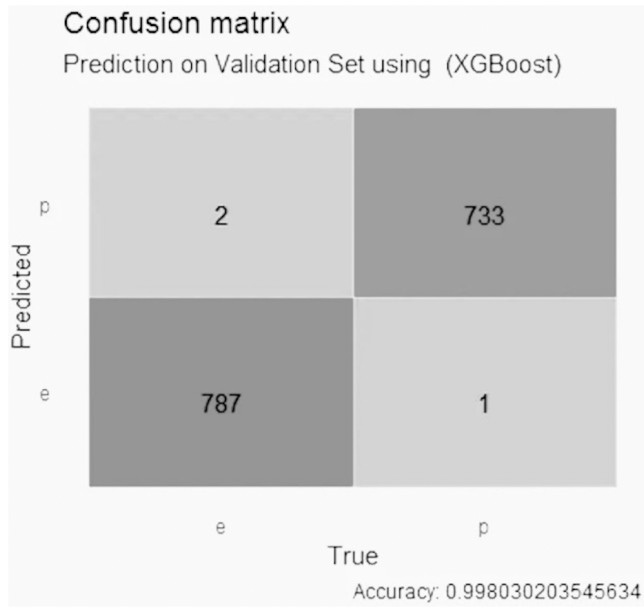

**Figure 10 Confusion matrix of the XGBoost model using GALGO selection.**

instances were not so favorable, they could not exceed more than 62%, that is why the proposed work aims to improve this prediction. Another interesting study that can be compared to the current research is the work by *Waqas, Baig & Ali (2009)*, which proposed the use of genetic algorithms on the same database used in this study. They applied Quin's

classification methodology to calculate the objective function, using the ID3 (Decision Tree) algorithm to classify the datasets, with two-thirds of the datasets used for training and the remainder for classification. Their study employed the non-dominated sorting genetic algorithm (NSGA) with each chromosome being binary and having a length equal to the total number of features. The results of their study showed the highest accuracies for toxicity (93.15) and edibility (91.4). The present study, on the other hand, used LASSO and GALGO feature selection methods and achieved similar results, indicating that these models are simpler than genetic algorithms and artificial neural networks, the number of features selected with GALGO was reduced to only 3, a point in favor over the aforementioned paper. Given that the use of 22 features is cumbersome in real-world applications, alternative proposals to reduce the number of features have been suggested. For example, "Behavioural features for mushroom classification" (*Ismail, Zainal & Mustapha, 2018*) identifies the most significant features for fungal classification. The selection of characteristics was performed by principal component analysis (PCA) using Waikato Environment for Knowledge Analysis (WEKA). The three most significant characteristics were odor with an average of 0.57, gill size 0.54, and bruises with 0.501. The odor variable appears as the most significant variable in the study, which is interesting given its agreement with LASSO and GALGO classifications. However, with these last two classifiers, the evaluation metrics are values higher than 90%, meaning that models that require only a single characteristic can determine the probability that a fungus is toxic. For the gill size variable, it is the second most important variable in our LASSO selection, but bruises are not significant in any of our models. Based on the results, the GALGO feature selection technique works better than the LASSO technique since it reduces the number of features to only three. Additionally, it is observed that XGBoost is the most suitable classification method. These findings provide insights into how feature selection techniques can improve the efficiency and accuracy of machine learning models for the classification of mushrooms. However, we acknowledge that the odor variable may require expertise and that it may be necessary to eliminate it in future work, if deemed necessary. In such cases, the remaining variables could still be used to classify mushrooms with good performance.

In 2019, *Alkronz et al. (2019)* used the same dataset and applied artificial neural networks (ANNs) models to classify fungi, using all 22 features. The ANNs model obtained a predictive accuracy of 99.25% for identifying the edibility of a mushroom. However, the results obtained from the three models with LASSO-selected variables in the present study were similar to those reported in the previous work, indicating that the models used in this study are much simpler than ANNs. Additionally, it should be noted that the use of simpler models can have practical advantages in terms of interpretability, computational efficiency, and ease of implementation. Moreover, the LASSO-selected variables could be useful for developing simpler models in real-world applications where there are limitations on computational resources. Therefore, the present study provides an alternative and practical method to classify mushrooms, with the potential to be implemented in practical scenarios. *Pinky, Islam & Alice (2019)* propose the use of bagging, boosting and random forest assembly methods for the classification of poisonous mushrooms and compare

the results to determine which algorithm achieves the best accuracy. In contrast to our work, they used five feature sets where each feature set contains the log2N+1 number of of attributes some of which are selected from the six features and rest attributes are selected randomly. The models obtained favorable results, accuracies close to 0.99, however, there are also models with 0.8 and in the worst cases 0.6 accuracy. It even mentions that dissimilarity measure-based bagging takes a longer time to show the result. It has taken almost 7 h for testing using all dataset, which is not favorable for obtaining an optimal prediction of a specimen. Unlike this work, our models maintain a lower computational cost, which will favor a faster prediction of toxicity. *Liu et al. (2022)* mentions the use of Deep Learning for shiitake mushroom grading using a high-efficiency channel pruning mechanism to improve the YOLOX method for mushroom identification and quality grading. The experimental results indicated that the improved YOLOX method proposed in this article can inspect the surface texture of shiitake mushrooms effectively, that mAP and FSP are respectively 99.96% and 57.3856, and the model size was reduced by more than half. This article provides an insight into the use of deep learning in mushroom classification, however, it is based on mushroom images, unlike the UCI database, these deep learning methods are useful for image recognition or large amounts of data. It is interesting to note that other approaches have proposed the use of all 22 features, making it difficult to implement in a real-world application.

In this study, using LASSO and GALGO feature selection techniques, a focused set of significant features for the classification of mushrooms' toxicity was determined. The consistent prominence of "odor" as a significant feature across different methodologies validates its importance, albeit its subjectivity in real-world scenarios. Compared to other research such as *Rashed Khan, Shajun Nisha & Sathik*'s (*2008*) use of expectation maximization (EM) and K-means, which could not surpass a 62% correctness rate, or *Waqas, Baig & Ali*'s (*2009*) genetic algorithm approach which achieved 93.15% accuracy for toxicity, the present study not only yielded comparable results but also reduced feature set complexity. Specifically, the GALGO feature selection technique excelled by narrowing the features down to three, outperforming other methods. Furthermore, while *Alkronz et al.* (*2019*) use of artificial neural networks (ANNs) attained a high accuracy of 99.25%, the models used in this study offer simplicity, interpretability, and computational efficiency. The utilization of simpler models, such as the ones highlighted in this research, presents an advantage, especially in real-world applications with computational constraints.

A considerable limitation of the data set is that it only contains specimens of the Agaricaceae family (*Agaricus* and *Lepiota* and allied species), which can be a problem for non- experts taxonomic users when encountering a specimen belonging to a different family. The obtained results are satisfactory, with classification models achieving more than 0.9 AUC. Furthermore, the reduction in the number of characteristics leads to improved identification of organisms and lower classification times and computational costs. The reduced number of features also simplifies the classification process for individuals, as they only need to analyze three characteristics, in the case of GALGO, to determine whether a fungus is toxic or edible. This study represents a significant contribution to addressing taxonomic issues related to fungi. The proposed approach simplifies the classification

process and minimizes the probability of making incorrect predictions, which reduces the risk of mushroom poisonings. This method facilitates the classification task and decreases the possibility of error. In future research, the aim is to transition from the current metrics, which are commonly recognized by campers or forest enthusiasts, to an advanced image classification system. Such a system would harness the capabilities of modern machine learning to visually identify and categorize mushrooms. This would not only enhance the accuracy of classification but also democratize knowledge by making it accessible to a wider audience. By doing so, laypersons unfamiliar with the intricacies of the fungal kingdom could confidently identify mushrooms, reducing the risk of potential mishaps and promoting a deeper appreciation for fungal biodiversity.

### Funding
The authors received no funding for this work.

### Competing Interests
The authors declare there are no competing interests.

### Author Contributions
- Jacqueline S. Ortiz-Letechipia conceived and designed the experiments, performed the experiments, analyzed the data, prepared figures and/or tables, and approved the final draft.
- Carlos E. Galvan-Tejada conceived and designed the experiments, performed the experiments, analyzed the data, authored or reviewed drafts of the article, and approved the final draft.
- Jorge I. Galván-Tejada analyzed the data, authored or reviewed drafts of the article, and approved the final draft.
- Manuel A. Soto-Murillo conceived and designed the experiments, performed the experiments, prepared figures and/or tables, and approved the final draft.
- Erika Acosta-Cruz analyzed the data, authored or reviewed drafts of the article, and approved the final draft.
- Hamurabi Gamboa-Rosales conceived and designed the experiments, authored or reviewed drafts of the article, and approved the final draft.
- José María Celaya Padilla performed the experiments, analyzed the data, authored or reviewed drafts of the article, and approved the final draft.
- Huizilopoztli Luna-García conceived and designed the experiments, prepared figures and/or tables, authored or reviewed drafts of the article, and approved the final draft.

### Data Availability
The data is available at UCI Machine Learning Repository: Mushroom. (1987). UCI Machine Learning Repository. https://doi.org/10.24432/C5959T.

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
