# Peer review of "Classification and selection of the main features for the identification of toxicity in Agaricus and Lepiota with machine learning algorithms"

_PeerJ, doi:10.7717/peerj.16501_

## Round 0.1 · original submission · Major Revisions

Reviewer 1 gives some general suggestions that should be readily met. More seriously, reviewer 2 notes that "working code is not shared. It is not clear from the article if any augmentation or preprocessing was done on the data. Some sample images for each stage of preprocessing if shared would make the article readable." which I agree with. Please make sure you address this critique. There were several other good suggestions to address.

·

Basic reporting

The manuscript is generally clear, unambiguous and understandable; a few instances where it could be clarified or improved are spelled out in detail below, under "additional comments". I would like a clear description of the mushrooms used in the dataset, which could be given as simply as "The datset encompasses the family Agaricaceae (Agaricus, Lepiota, and allied species) as described in Lincoff (1981) The Audubon Society Field Guide to North American Mushrooms." I was able to determine this by going to the referenced Center for Machine Leraning and Intelligent Systems and searching for "mushrooms" but providing the actual source will be more convenient for users.

Experimental design

I have some questions about how the findings reported in this paper might be implemented. As I read it, the point of the study to some extent is to identify the most useful features in predicting mushroom toxicity, such that users (presumably not taxonomic experts) can focus on the more important features – a worthy goal. However, a non-expert may very well not know whether the mushroom in front of them is an Agaricus or Lepiota (on which these algorithms are trained), or, say, an Amanita or a Tricholoma. Odor is not likely to be an especially useful character in identifying deadly Amanita species, while gill attachment, ring type, and “stalk root” likely are. I see no trouble with confining this paper to Agaricus and Lepiota – you have to start somewhere! – but somewhere the limitations need to be acknowledged.

Validity of the findings

I have some questions about how the findings reported in this paper might be implemented. As I read it, the point of the study to some extent is to identify the most useful features in predicting mushroom toxicity, such that users (presumably not taxonomic experts) can focus on the more important features – a worthy goal. However, a non-expert may very well not know whether the mushroom in front of them is an Agaricus or Lepiota (on which these algorithms are trained), or, say, an Amanita or a Tricholoma. Odor is not likely to be an especially useful character in identifying deadly Amanita species, while gill attachment, ring type, and “stalk root” likely are. I see no trouble with confining this paper to Agaricus and Lepiota – you have to start somewhere! – but somewhere the limitations need to be acknowledged.

Additional comments

Line 32: suggest “which includes organisms commonly referred to as mushrooms”, as the majority of organisms in kingdom fungi would not be considered mushrooms (yeasts, molds, etc.)
Line 45: replace “mycotoxins” with “toxins”; “mycotoxins” is usually treated as a distinct term used to refer specifically to toxins produced by microfungi (molds) in food or feed (examples being aflatoxins, trichothecenes, ergot alkaloids, etc.)
Lines 45-49: “Ingesting poisonous mushrooms... ...which are often fatal” is pretty much a direct translation of Lazo’s (1986) text into English. Several of these terms are not the ones used in English (and “gyromitrpsychic” is a typo, while “parafaloidínico” is a term I find only in the Spanish language literature, and appears to refer to poisoning by Lepiota locaniensis, a species presumed extinct); suggest using instead the terminology outlined in White J, Weinstein SA, De Haro L, Bédry R, Schaper A, Rumack BH, Zilker T (2018) Mushroom poisoning: A proposed new clinical classification. Toxicon. https://doi.org/10.1016/j.toxicon.2018.11.007. This would mainly involve replacing “phalloidin and paraphalloidin” with “cytotoxic (including amatoxins)” (and “gyromitrpsychic” with “gyromitric”)
Line 104 and elsewhere: please be clear in the terminology and what is being described. In the taxonomic sense, Agaricus and Lepiota belong to the same family (Agaricaceae). You could use a narrow circumscription and say “in the genera Agaricus and Lepiota”; alternately, as you are including species of Lepiota sensu lato that are no longer classified in Lepiota (such as the green-gilled, poisonous Chlorophyllum molybdites, it would be better to say “in the family Agaricaceae”.
Figure 2: “space.print.c9l9r” presumably should be “spore.print.color”. Also check spelling of “color” for this feature in Figures 3 and 4.

·

Basic reporting

The language usage seems to be good.
 The article did not include sufficient references related to the work done or the area of study. General materials, books, and articles were used for reference which show that the authors have not aware of recent trends in the field of study.
 Figures seem to be good enough with clarity even then the confusion metric, and classification performance metrics obtained by the code if shared as images would be interesting.
 The raw data is shared but still, the working code is not shared. It is not clear from the article if any augmentation or preprocessing was done on the data. Some sample images for each stage of preprocessing if shared would make the article readable.
 Further analysis has to be made including some predictive results on the classification done like prediction score along with image proofs, and confusion matrix on classification, instead of moving with very basic machine learning algorithms some deep learning techniques would have also been included.
 If the authors are concentrating only on the machine learning algorithms, then the updated methods or hybrid methods with some existing genetic algorithm or optimization algorithms would be preferable to show the work more innovative.
 As the authors are aware that R or Python implementations of the algorithms have inbuilt methods with the possibility to vary the parameter and tune the model. Even one single method to show the results of all available machine learning algorithms like “lazy learning” are available, so no innovativeness could be observed from the article.

Experimental design

The research objectives are not well-defined, they should be defined and projected at the end of section 1.
 The research gaps identified should be submitted at the end of section 2 after the related works are analysed. Since the Related works (State of art) are not well-defined in the article, we believe that the author will not be able to find the research gap.
 76 works have cited the used dataset, but no such referring methods were utilized for comparative analysis.
 The reason for choosing 2 methods for feature selection and 3 methods for classification is not justified.
 Methods chosen for analysis should be justified why they are chosen and taken up for study.
 Further investigations as suggested are needed to make the article innovative.
 The performance metrics chosen are to be defined to enable readability in budding researchers.

Validity of the findings

Qualitative results are provided in Tables 2 and 3, but image-oriented results on classification based on score analysis would give the author a better understandability and readability.
 Table 1 shows a repetition of the original page on UCI for Dataset description, which is no further needed in the article since the reference to the data page is provided.
 Analysis of data used for classification (count on the classes) should be specified.
 The dataset seems to be unbalanced and has missing values. How were they dealt with?
 Comparative analysis of results obtained from previously published works is to be included.
 Only after comparative analysis, a conclusion can be obtained that the proposed method is quite good enough. Note. No such proposed work exists.

Additional comments

No proposed work could be found. Hybrid methods based on genetic algorithms or optimization algorithms could be adopted.
 Existing methods (conventional) were chosen.
 Comparative analysis of the results with existing works should be included to prove a proposed work is innovative.
 Exploratory data analysis and preprocessing of data are missing
 Working code is not found
 Image classification article should include some results based on images
 Performance metrics including prediction scores on the classified mushroom have to be found by code.
 The article needs enormous effort and update to be accepted for publication in a journal

---

## Round 0.2 · Minor Revisions

The reviewers have a few minor revisions that I suggest you follow. There will be no need to send any minor revision you make back to the reviewers.

·

Basic reporting

The authors have addressed my concerns from the previous manuscript. The English is clear, professional, and understandable. I will leave the detailed review of the computational science to Reviewer 2, as it is not within my scope.

To my mind, Figure 2 does not add useful information and is, if anything, confusing the matter. The statement in line 106, “The dataset is well-balanced, with 52% of the mushrooms being poisonous and 48% being edible.” Is sufficient; the reader’s eye is not going to be able to tell that “e” occupies 52% of the figure while “p” occupies 48% (as opposed to 50:50, 54:46, etc.). The line “gray segments are missing values” in the figure is meaningless (there are no gray segments”, and the use of “e”, “p” and “categorical values” are needlessly vague; if the figure is to be included, there is plenty of room to spell out “edible” and “poisonous”. But I would argue against its inclusion.

Line 152: should that be “KNN as a ‘lazy learning’ classification model” (as written), or “KNN is a ‘lazy learning’ classification model”?

Experimental design

No comment

Validity of the findings

No comment

·

Basic reporting

No comment

Experimental design

Eventhough the rebuttal is quite convincing, I would suggest to add some images self labelled by the machine leanring algorithm in classification regarding the actual class and predicted class or the percentage of accuracy on prediction for each sample image. Atleast 5 images from dataset should be shown up.

Validity of the findings

No comment

Additional comments

The authors seems to convince the reviewer rather than incorporating the changes requested completely. Most of the comments were addressed, Happy for that but still comment 1,3,4,5(objective is not just classification it should be optimized and convince the other methods existing in the state of art) and 8 were not addressed.

---

## Round 0.3 · Minor Revisions

Hi, I do suggest you change Lines 346-347 as suggested by reviewer 1. I do not have to send back to reviewers, Sincerely,

·

Basic reporting

No comment

Experimental design

No comment

Validity of the findings

No comment

Additional comments

Lines 346-347: "serious health effects", in this context, presumably refers to the risk of mushroom poisoning; "homoplasies, phenotypic plasticity, and hybrids" may lead to "erroneous predictions", but are not the result thereof. This single sentence needs rewriting.

·

Basic reporting

Well done

Experimental design

Sufficient explanations are provided and required changed are made as suggested

Validity of the findings

Convincing

Additional comments

Acceptable article

---

## Round 0.4 · accepted · Accept

Thank you for replying to the reviewers.